# Blood group typing from whole-genome sequencing data

**Julien Paganini**[1], **Peter L. Nagy**[2¤], **Nicholas Rouse**[2], **Philippe Gouret**[1], **Jacques Chiaroni**[3], **Chistophe Picard**[3], **Julie Di Cristofaro**[3]*

**1** Xegen, Gemenos, France, **2** Laboratory of Personalized Genomic Medicine, Columbia University, New York, New York, United States of America, **3** Aix Marseille Univ, CNRS, EFS, ADES, "Biologie des Groupes Sanguins", Marseille, France

¤ Current address: Praxis Genomics LLC, Atlanta, Georgia, United States of America
* julie.dicristofaro@efs.sante.fr

**Data Availability Statement:** All relevant data are within the manuscript and its Supporting Information files. Sequencing data are available at http://www.ncbi.nlm.nih.gov/bioproject/662371.

## Abstract

Many questions can be explored thanks to whole-genome data. The aim of this study was to overcome their main limits, software availability and database accuracy, and estimate the feasibility of red blood cell (RBC) antigen typing from whole-genome sequencing (WGS) data. We analyzed whole-genome data from 79 individuals for HLA-DRB1 and 9 RBC antigens. Whole-genome sequencing data was analyzed with software allowing phasing of variable positions to define alleles or haplotypes and validated for HLA typing from next-generation sequencing data. A dedicated database was set up with 1648 variable positions analyzed in KEL (KEL), ACKR1 (FY), SLC14A1 (JK), ACHE (YT), ART4 (DO), AQP1 (CO), CD44 (IN), SLC4A1 (DI) and ICAM4 (LW). Whole-genome sequencing typing was compared to that previously obtained by amplicon-based monoallelic sequencing and by SNaP-shot analysis. Whole-genome sequencing data were also explored for other alleles. Our results showed 93% of concordance for blood group polymorphisms and 91% for HLA-DRB1. Incorrect typing and unresolved results confirm that WGS should be considered reliable with read depths strictly above 15x. Our results supported that RBC antigen typing from WGS is feasible but requires improvements in read depth for SNV polymorphisms typing accuracy. We also showed the potential for WGS in screening donors with rare blood antigens, such as weak JK alleles. The development of WGS analysis in immunogenetics laboratories would offer personalized care in the management of RBC disorders.

## Introduction

Whole-genome data has become more accessible thanks to techniques being made easier, the availability of sequencing machines or contractors, and the release of public data. Only a small part of these entire genomes are exploited beyond the scope of their initial purposes.

Amplicon-based next-generation sequencing (NGS) assays have in many ways laid the groundwork for whole-genome analyses as they require equivalent reagents, equipment and experimental skills. Much software for amplicon-based NGS has been developed, validated and certified in clinical fields. More particularly, most immunogenetics labs are equipped with

**Funding:** No funding was received for this research. The funder provided support in the form of salaries for authors JP, PG and PN, but did not have any additional role in the study design, data collection and analysis, decision to publish, or preparation of the manuscript. The specific roles of these authors are detailed in the 'author contributions' section. The authors received no specific funding for this work. Authors Julien Paganini and Philippe Gouret are employed by a commercial company: Xegen, Gemenos, France. Author Peter L. Nagy is employed by a commercial company: Praxis Genomics LLC, Atlanta, Georgia, USA.

**Competing interests:** The authors have no conflicts of interest to declare. Commercial affiliation of JP, PG and PN does not alter our adherence to all PLOS ONE policies on sharing data and materials.

amplicon-based NGS for HLA typing and some have also developed and validated such techniques for human platelet and RBC antigens [1–3].

Many questions can be explored in various fields thanks to WGS resources and their integrative investigation; first in population genetics where such data may improve understanding of natural selection, local adaptation, demographic history and early human migration [4,5]. Then in evolutionary genetics where it can address more fundamental issues such as gene evolution and functional investigation [4,5] thanks to haplotype reconstruction or the localization of new variants. Finally, these rapidly evolving techniques have now made their entry into analysis on an individual scale, for example in forensics and for clinical purposes [3,6].

However many issues raised by WGS handling limit the implementation of these techniques both in Research and Clinical laboratories working within regulatory approved frameworks e.g. Council of Europe (CE): software availability, database accuracy and editing, coverage and read depth quality indicators [5,7]. Thus, many whole-genome experiments designed for one scientific purpose are not used for any further analyses.

The aim of this study was to overcome these limitations and to estimate the feasibility of RBC antigen typing from WGS data. We analyzed whole-genome data from 79 individuals from Central Asia [8] for the highly polymorphic *HLA-DRB1* gene and for 9 blood group antigen.

The same samples had previously been typed for HLA-DRB1 by amplicon-based monoallelic sequencing and for blood group bi-allelic polymorphisms using SNaPshot analysis [9].

WGS data was analyzed with software validated for HLA typing from NGS data [10]. This software relies on an allele alignments database; whereas the HLA system has a very convenient database and consensus on allele naming [11] with monthly updates, genetic polymorphism of RBC antigens are provided in Portable Document Format (pdf) and need to be converted. Most importantly, the software used in this study allows phasing of variable positions to define alleles or haplotypes. In a second analysis, the software was set up to search WGS data for new alleles. Indeed, previous investigations for blood groups but also for specific anthropogenic analyses revealed that this cohort presented a singular genetic mosaic of components from various geographic regions of Eurasian ancestry [12].

## Materials and methods

### DNA samples

Seventy-nine samples were used in this study formerly analyzed for anthropogenic markers and described in [12]. All samples were obtained from unrelated male Afghan volunteers after obtaining written informed consent. The study protocol was registered by the Ministere de l'Enseignement Superieur et de la Recherche in France (committee 208C06, decision AC-2008-232). Institutional review board Ministere de l'Enseignement Superieur et de la Recherche in France committee 208C06, (decision AC-2008-232) specifically approved this study.

### Blood group genotyping by SNaPshot analysis

Samples were analyzed for main RBC antigens and results have been previously published [9]. DNA was genotyped for the Kell (KEL), Duffy (FY), Kidd (JK), Cartwright (YT), Dombrock (DO), Indian (IN), Colton (CO), Diego (DI) and Landsteiner-Wiener (LW) systems by SNaPshot analysis (corresponding genes according to ISBT nomenclature: *KEL*, *ACKR1* (FY), *SLC14A1* (JK), *ACHE* (YT), *ART4* (DO), *AQP1* (CO), *CD44* (IN), *SLC4A1* (DI) and *ICAM4* (LW) [13]. Determination of blood group antigens, other than those of the ABO, RH and MNS systems, depends mainly on the presence of one or more SNPs in the coding sequence. Fourteen SNPs were analyzed corresponding to bi-allelic polymorphism (KEL p.Thr 193Met

(KEL:1,-2), KEL p.Leu597Pro (KEL:6,-7), FY p.Gly42Asp (FY:2), FY p.Arg89Cys (Fya+w), FY c.-67T>C (Fy(a-b-) erythroid cells only), JK p.Asp280Asn (JK:2), YT p.His353Asn (YT:-1,2), DO p.Asn265Asp (DO:2), DO p.Gly108Val (DO:-4), DO p.Thr117Ile (DO:-5), IN p.Arg46Pro (IN:1,-2), CO p.Ala45Val (CO:2), DI p.Pro854Leu (DI:1,-2) and LW p.Gln100Arg (LW:7) (https://www.isbtweb.org).

### *HLA-DRB1* typing by monoallelic sequencing

*HLA-DRB1* was typed by monoallelic sequencing using Protrans HLA SBT S3 (Protrans) according to manufacturer's instructions. This kit relies on locus specific amplification followed by monoallelic Sanger sequencing.

### Whole-genome NGS library preparation and data acquisition

Detailed description of WGS procedure is given in [8]. DNA samples were sonicated using a Covaris S220 Ultrasonicator to yield fragments with a median fragment length of 300 bps according to the manufacturer's recommendations. Low molecular weight DNA (<300 bps) enrichment from all samples was performed using AMPure XP beads (NEB). The library was prepared using the TruSeq Nano DNA LT kit (Illumina) according to the manufacturer's recommendations. Library size and quality was confirmed with Fragment Analyzer (Advanced Analytical) and quantitative PCR (Biorad S1000; CFX96 Real Time System). Paired-end sequencing (2x150 bps) was performed on the Illumina NovaSeq 6000 System (Illumina) following the manufacturer's recommendations.

### Whole genome data analysis

**Pre-alignment processing.** Demultiplexing of runs was performed in BaseSpace (www.illumina.com/BaseSpaceApps). Prior to analysis, quality and adapter trimming was performed by Trim Galore (Babraham Bioinformatics http://www.bioinformatics.babraham.ac.uk/projects/trim_galsore/) on all fastq files from all runs. Low quality bases with a Phred score below 20 (Q20) were removed from the 3 prime end of the reads followed by the removal of any Illumina adapter contamination (minimum adapter match of 3 with an allowed matching error rate of 0.1). Reads of less than 40 after quality and adapter trimming were removed and only properly paired-end read data were retained and analyzed.

**Sequencing data quality assessment.** Sequencing performance relies mainly on genome coverage and read depth [14]. WGS data quality was assessed by the quantity of reads obtained per sample. Mean read depth of genome was estimated for each sample by the total number of reads X read size [150 bps] / genome size [2,867,437,753 bps]. Mean read depth for each gene was also estimated by the number of reads mapped X read size [150 bp] / gene size.

**Statistical analyses.** Statistical analyses were performed with GRAPH PAD Prism 5 software (California USA, www.graphpad.com). Number of reads are presented as mean and range [min, max]. Differences among number of reads according to typing gene status were tested using Kruskal-Wallis one-way ANOVA for three values and Mann Whitney test for two values. Threshold for significance (alpha) was set at 0.05.

**Blood group typing and HLA-DRB1 allelic assignment.** PolyPheMe software (Xegen, France) was used to perform all typing from WGS data. WGS data were directly aligned to each gene as reference, no human genome was used for read mapping. Alignments were generated by PolyPheMe software with a Bowtie tool [15,16].

Genetic polymorphisms of RBC antigens described in [3] and *International Society of Blood Transfusion* (ISBT; http://www.isbtweb.org) were used for genetic alignment construction. Reference alleles were generated for *KEL*, *ACKR1* (FY), *SLC14A1* (JK), *ACHE* (YT), *ART4*

(DO), *AQP1* (CO), *CD44* (IN), *SLC4A1* (DI) and *ICAM4* (LW) genes. The other blood group database for which updates were stopped in 2017 was not used for this study [17,18]. 1648 variable positions (68 for *KEL*, 228 for *FY*, 904 for *SLC14A1*, 4 for *ACHE*, 21 for *ART4*, 387 for *CO*, 5 for *IN*, 20 for *DI* and 11 for *LW* were analyzed with PolyPheMe v1.2 on WGS data. All positions analyzed and their corresponding alleles are given in S1 Table. A minimum threshold was defined at 5 reads per position analyzed. The PolyPheMe software can phase heterozygous positions and identify haplotypes when reads overlap. WGS analysis validation was based on a comparison with the 14 positions described by SNaPshot assays.

In a second phase, potential new alleles were estimated by previously unidentified combinations of known polymorphisms but also by polymorphisms unmapped in the ISBT database. For these unreported alleles, WGS data was re-analyzed and polymorphisms were taken into consideration if they had a minimum threshold of 10 reads per position combined with a minimum of 5 occurrences.

*HLA-DRB1* was typed at second field resolution with specific parameters for HLA systems previously described [10] using the IMGT 3.39.0 database [11] as reference. This analysis used allele typing according to polymorphisms described in the database. A second analysis was performed on WGS data to find potential new polymorphisms.

## Results

### Sequencing data quality

Sequencing data are available at http://www.ncbi.nlm.nih.gov/bioproject/662371. Genome sequencing displayed a mean of 34 Gb [16–53].The mean read depth of the genome, estimated for each sample by the total number of reads X read size / genome size, was 11.8x [5.5x-18.4x]. Mean read depth for each gene, estimated by the number of reads mapped X read size [150 pb] / gene size, are given in S2 Table.

### Blood group analyses

Blood group genotyping analyzed by SNaPshot are given in Table 1. Most analyses focused on one SNP leading to bi-allelic results, except for KEL, DO and FY systems for which 2 or 3 SNPs were analyzed. Most antigens displayed low or no allelic diversity except for DO (p. Asn265Asp), FY (p.Gly42Asp), JK (p.Asp280Asn) and YT (p.His353Asn).

Group typing based on WGS analysis was performed targeting all of the variable positions described in S1 Table. Sixty-three alleles out of 1035 described by SNaPshot could not be resolved (6.1%) by WGS analysis. For all genes analyzed, typing resolution was associated with the number of reads mapped on their genetic sequence (S3 Table).

WGS-based typing showed 100% of concordance for homozygous SNPs analyzed by SNaPshot (N = 865 SNPs) and 95.3% for heterozygous positions (N = 102/107 SNPs; Table 2).

98.6% of WGS-based typing results were concordant with SNaPshot results for KEL (p. Met193Thr), one heterozygous sample was not correctly typed (KEL*02) and 5 samples remained unresolved. The monomorphic position KEL (p.Pro597Leu) was 100% concordant, 3 samples were unresolved.

100% of concordance was observed for the monomorphic positions FY (p.Arg89Cys) and FY -67T>C, 4 and 8 samples remained unresolved respectively.

98.4% of WGS-based typing results were concordant with SNaPshot results for FY (p. Gly42Asp); among the 30 heterozygous samples, 1 was typed FY*01. Six samples were not resolved.

98.6% of WGS-based results were concordant with SNaPshot results for JK (p.Asp280Asn), with 1 incorrect typing for a heterozygous sample (JK*02). Eight samples were not resolved.

**Table 1. Blood group typing by SNaPshot analysis.**

| Polymorphism | Allele | Ho wt | Ho mt | He | ND |
|---|---|---|---|---|---|
| KEL (578C>T) | KEL*02 | 75 | | 4 | |
| KEL (1790T>C) | KEL*02.06 | 72 | | | 7 |
| ACKR1 (-67T>C) | FY*01N.01 | 72 | | | 7 |
| ACKR1 (125G>A) | FY*02 | 19 | 16 | 30 | 14 |
| ACKR1 (265C>T) | FY*01W.01 | 65 | | | 14 |
| SLC14A1 (838G>A) | JK*01 | 17 | 24 | 37 | 1 |
| ACHE (1057C>A) | YT*02 | 68 | | 11 | |
| ART4 (793A>G) | DO*02 | 12 | 27 | 33 | 7 |
| ART4 (323G>T) | DO*02.–04 | 72 | | | 7 |
| ART4 (350C>T) | DO*01.–05 | 79 | | | |
| CD44 (137G>C) | IN*01 | 72 | | | 7 |
| AQP1 (134C>T) | CO*02 | 79 | | | |
| SLC4A1 (2561C>T) | DI*02 | 77 | | 2 | |
| ICAM4 (299A>G) | LW*07 | 72 | | | 7 |
| Total | | 851 | 67 | 117 | 71 |

Blood groups genotyped by SNaPshot analysis (Ho wt: Homozygous wild type, Ho mt: Homozygous mutated, He: Heterozygous, ND: Not defined).

100% of concordance was observed for the monomorphic positions DO (p.Gly108Val) and DO (p.Thr117Ile); 4 and 3 samples remained unresolved respectively. 98.6% of WGS-based results were concordant with SNaPshot results for DO (p.Asn265Asp), 1 heterozygous sample was incorrectly typed (DO*02). Four samples were not resolved.

100% of concordance was observed for YT (p.His353Asn) (9 samples were unresolved), IN (p.Pro46Arg) (2 samples were not resolved), CO (p.Ala45Val) (4 samples were not resolved,) and LW (p.Gln100Arg) (1 sample not resolved).

**Table 2. Blood group typing by WGS analysis.**

| Gene | Polymorphism | Homozygous positions | | | Heterozygous positions | | |
|---|---|---|---|---|---|---|---|
| | | Correct typing | Incorrect typing | ND | Correct typing | Incorrect typing | ND |
| KEL | KEL (578C>T) | 70 | | 5 | 2 | 1 | 1 |
| | KEL (1790T>C) | 69 | | 3 | | | |
| ACKR1 (FY) | ACKR1 (-67T>C) | 64 | | 8 | | | |
| | ACKR1 (125G>A) | 32 | | 3 | 26 | 1 | 3 |
| | ACKR1 (265C>T) | 61 | | 4 | | | |
| SLC14A1 (JK) | SLC14A1 (838G>A) | 37 | | 4 | 32 | 1 | 4 |
| ACHE (YT) | ACHE (1057C>A) | 60 | | 8 | 10 | | 1 |
| ART4 (DO) | ART4 (793A>G) | 36 | | 3 | 31 | 1 | 1 |
| | ART4 (323G>T) | 68 | | 4 | | | |
| | ART4 (350C>T) | 76 | | 3 | | | |
| CD44 (IN) | CD44 (137G>C) | 70 | | 2 | | | |
| AQP1 (CO) | AQP1 (134C>T) | 75 | | 4 | | | |
| SLC4A1 (DI) | SLC4A1 (2561C>T) | 76 | | 1 | 1 | 1 | |
| ICAM4 (LW) | ICAM4 (299A>G) | 71 | | 1 | | | |
| Total | | 865 | | 53 | 102 | 5 | 10 |

Validation of WGS-based blood group typing according to SNaPshot results (N: Number, ND: Not defined, unresolved).

98.7% of concordance was observed for DI (p.Pro854Leu) with 1 incorrect typing for a heterozygous sample (DI*02); 2 samples remained unresolved.

WGS-based typing targeting all of the variable positions (described in S1 Table) led to ambiguities (described in S4 Table) but also to more precise typing. WGS analysis allowed typing of JK*01W.01 allele corresponding to JK:1^WK phenotype in 28 samples [19]; 10 samples were JK*02/JK*01W.01, 6 were JK*01/JK*01W.01 and 1 sample was homozygous for JK*01W.01. FY*02 allele associated with c.298G>A (p.Ala100Thr) was found in 18 samples [20]. No SNaPshot results were available to confirm or refute these typing results.

WGS data analysis also revealed polymorphisms that were unmapped in the ISBT database. A total of 267 previously unidentified polymorphisms covered with a minimum depth of 10x and observed in a minimum of 5 samples were found (S5 Table). Among these, 5 SNPs were in exonic regions but none led to amino-acid changes. Two SNPs in the *DO* gene were observed in 18 and 21 samples, 2 SNPs in *IN* were observed in respectively 37 and 41 samples and, in the *JK* gene, one SNP was observed in 37 individuals (S6 Table).

## HLA-DRB1 analyses

Thirty-four *HLA-DRB1* alleles were defined at maximum resolution by amplicon-based monoallelic sequencing, 5 samples could not be analyzed (Table 3).

Ninety-one percent of WGS-based *HLA-DRB1* typing, i.e. 135 out of 148 alleles, showed an exact match with typing defined by monoallelic sequencing at second field resolution. Most discordances were due to insufficient coverage and low read numbers leading to differences in 3rd and 4th digits; two samples (counting for 4 alleles) could not be typed.

No novel polymorphism could be detected in HLA-DRB1 during the second analysis of the WGS data.

## Discussion

In this study we explored diploid markers in WGS data generated for Y-chromosome analysis from 79 individuals [8]. Analyses were performed with Polypheme software validated for HLA typing from NGS data [10] and set up for RBC analysis. *HLA-DRB1* gene and 9 blood group antigens were typed (*KEL*, *ACKR1* (FY), *SLC14A1* (JK), *ACHE* (YT), *ART4* (DO), *AQP1* (CO), *CD44* (IN), *SLC4A1* (DI) and *ICAM4* (LW)) according to standard nomenclature (IMGT 3.39.0 database [11], ISBT (http://www.isbtweb.org) and RBC antigens [3]). Whereas targeted strategies, such as PCR followed by sequencing or SnaPshot, circumvent specificity issues of genes with structural changes and hybrids such as *RHCE/RHD* and *GPA/GPB*; their analysis from WGS data have requires specific bioinformatic approaches including CNV (copy number variation) analysis. Therefore, such systems were not included in this study.

Our results showed that blood group typing deduced from WGS were correct at 99.5% compared to SNaPshot analysis (967 SNP correctly identified out of 972 typed); 93% when taking into account ambiguous typing. In a clinical or research context however, ambiguous RBC results need to be reanalyzed. *HLA-DRB1* typing from WGS showed 91% of concordance with those obtained by amplicon-based monoallelic sequencing. These performances on RBC antigens were similar to those presented in a former study on WGS from donor data [3] which included the typing of highly complex genes such as MNS, RHD/RHCE and ABO systems.

WGS data quality is assessed by the estimation of read depth. A former study conducted on WGS data established a minimum of 15x for RBC antigen typing in the clinical field [3,14]. Here, mean read depth of the genome was estimated at 11.8x [5.5x-18.4x] and read depth for each gene reached higher values. For each gene, typing resolution was significantly associated with the number of reads mapped on its sequence and ambiguous and incorrect typing showed

**Table 3. HLA-DRB1 analysis by monoallelic sequencing.**

| HLA-DRB1 | No. |
|---|---|
| *07:01:01 | 16 |
| *11:04:01 | 16 |
| *03:01:01 | 15 |
| *13:01:01 | 13 |
| *01:01:01 | 11 |
| *15:01:01 | 11 |
| *14:01:01/14:54 | 9 |
| *15:02:01 | 8 |
| *11:01:01/11:01:08 | 7 |
| *10:01:01 | 5 |
| *01:02:01 | 3 |
| *04:01:01 | 3 |
| *11:03 | 3 |
| *15:06 | 3 |
| *16:02:01 | 3 |
| *04:03:01 | 2 |
| *04:04:01 | 2 |
| *04:05:01 | 2 |
| *11:01:01 | 2 |
| *14:04 | 2 |
| *04:01:03 | 1 |
| *04:02 | 1 |
| *07:05 | 1 |
| *08:01:01/08:01:03 | 1 |
| *08:03:02 | 1 |
| *11:42 | 1 |
| *12:01:01/12:06/12:10/12:17 | 1 |
| *12:02:01 | 1 |
| *13:02:01 | 1 |
| *13:03:01 | 1 |
| *14:01:03 | 1 |
| *14:07:01 | 1 |
| *14:10 | 1 |
| *15:02:02 | 1 |
| ND | 10 |

HLA-DRB1 allelic typing results using monoallelic sequencing (No.: Number of alleles; ND: Not defined).

low numbers of reads corresponding to the missing allele and read depth equal to or below 15x. Our study thus confirms that RBC typing from WGS should be considered reliable with read depths strictly above 15x. To reach this goal, genome sequencing of one human (3Gb) should be analyzed with at least 45 Gb of data, here mean data was 34 Gb [16–53].

In our study, WGS data analysis allowed refined typing, identification of both potential new alleles and haplotypes as PolyPheMe software used here allowed phasing of polymorphisms subject to sufficient coverage and variable positions. We were able to type the JK*01W.01 allele [19] and the FY*02 allele associated with c.298G>A [20]. The weak JK allele may present a risk of hemolytic transfusion reactions [21] as it has been shown that among

samples screened as JK:-1,-2, a fraction was JK:1$^{WK}$ [22]. JK*01W.01 has been reported in Caucasian, Asian and Chinese individuals [19] but there is a lack of description of this allele among different populations. Given the frequency found here, our results strongly support the need of a better description of this allele, particularly in Asia.

Serological typing is the gold standard for blood group analysis but in particular situations molecular analysis can provide valuable information. In hematology laboratories, molecular biology based on sequence analysis was superseded by ready-to-use closed systems mainly based on SNPs analysis and validated for clinical purposes. Whereas unthinkable for routine patient care, some situations would gain from WGS such as screening donors for rare blood antigens and the management of RBC disorders. In this regards, our results showing rare and potential new alleles are particularly relevant in diseases such as Sickle Cell disease for example, where allo-immunization is a major complication [23]. Research of minor alleles and their potential role in allo-immunization in these patients would be a major advance in personalized medicine.

In a second analysis, WGS data were screened for new polymorphisms. 262 new variable positions in intronic regions and 5 polymorphisms in exons were identified, none led to non-synonymous mutations. An insight of their frequencies in populations described as being related to the Afghan population would contribute to refining their origins [12].

Molecular testing for the HLA system has been integrated in immunogenetics laboratories for a long time and evolves according to new technologies. Amplicon-based NGS is suitable for donor HLA typing, with robust and certified protocols, high throughput and highly resolutive typing results. These protocols can be performed with methods requiring several days and are also suitable for patients, for whom typing results are rarely impatiently awaited.

Immunogenetics laboratories are thus quite prepared to integrate WGS in their pipeline and use it to analyze other immune markers. Patients with auto-immune diseases, solid organ and HSC transplantation, or inflammatory diseases would benefit from personalized care with specific typing of non-classical HLA, FC receptors, KIR or LILRs [24–27]. In conclusion, the implementation of WGS can serve many purposes, from anthropogenic integrative studies to handling specific diseases in clinical fields.

## Supporting information

**S1 Table. Positions analyzed in blood group genes.** Positions analyzed for blood group typing and their corresponding alleles.
(XLSX)

**S2 Table. Number of reads and read depth.** Mean [min-max] number of reads and estimated read depth for each blood group gene analyzed. For each locus, gene size and effective size (i.e. sequence without repeated patterns in intronic sequences) are given.
(DOCX)

**S3 Table. Typing resolution and number of reads.** Typing status according to number of reads (mean [min-max]) (No.: Number) and read depth (mean [min-max]); (Incorrectly typed samples could not be included in the statistical analysis (N = 1)).
(DOCX)

**S4 Table. Typing ambiguities.** WGS blood group results ambiguities (No.: Number).
(DOCX)

**S5 Table. Unreported polymorphisms.** Number of polymorphisms revealed by whole-genome analysis but not described in the ISBT database (observed in at least 5 samples with a

minimum coverage of 10 reads).
(DOCX)

**S6 Table. New polymorphisms in exons.** Description of exonic SNPs revealed by whole-genome analysis. Note that mutations in IN are located after the codon stop (exon 9) in IN isoform 4 described in ISTB.
(DOCX)

## Author Contributions

**Conceptualization:** Julien Paganini, Jacques Chiaroni, Julie Di Cristofaro.

**Data curation:** Julien Paganini, Peter L. Nagy, Nicholas Rouse, Philippe Gouret, Julie Di Cristofaro.

**Formal analysis:** Julien Paganini, Nicholas Rouse, Julie Di Cristofaro.

**Investigation:** Jacques Chiaroni, Julie Di Cristofaro.

**Methodology:** Julien Paganini, Peter L. Nagy, Nicholas Rouse, Chistophe Picard, Julie Di Cristofaro.

**Project administration:** Jacques Chiaroni, Julie Di Cristofaro.

**Resources:** Peter L. Nagy.

**Software:** Julien Paganini, Philippe Gouret.

**Supervision:** Julie Di Cristofaro.

**Validation:** Julien Paganini, Peter L. Nagy, Jacques Chiaroni, Chistophe Picard, Julie Di Cristofaro.

**Writing – original draft:** Julien Paganini, Julie Di Cristofaro.

**Writing – review & editing:** Peter L. Nagy, Nicholas Rouse, Philippe Gouret, Jacques Chiaroni, Chistophe Picard.

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
