## [Decision Letter · Decision Letter 0]

6 Aug 2020

PONE-D-20-17434

BLOOD GROUP TYPING FROM WHOLE-GENOME SEQUENCING DATA

PLOS ONE

Dear Dr. Di Cristofaro,

Thank you for submitting your manuscript to PLOS ONE. I apologize for the long time that it has taken for the review process. I have received comments from two reviewers so far, and, after carefully considering them, feel that while the manuscript has merit, it does not fully meet PLOS ONE’s publication criteria as it currently stands. Therefore, I invite you to submit a revised version of the manuscript that addresses all the points that have been raised by the reviewers.

We look forward to receiving your revised manuscript.

Kind regards,

Santosh K. Patnaik, MD, PhD

Academic Editor

PLOS ONE

Journal Requirements:

2. Please provide additional details regarding participant consent. In the ethics statement in the Methods and online submission information, please clarify what type of consent you obtained (for instance, written or verbal, and if verbal, how it was documented and witnessed).

3. To comply with PLOS ONE submission guidelines, in your Methods section, please provide additional information regarding your statistical analyses. For more information on PLOS ONE's expectations for statistical reporting, please see https://journals.plos.org/plosone/s/submission-guidelines.#loc-statistical-reporting.

4.Thank you for stating the following in the Financial Disclosure section:

[The author(s) received no specific funding for this work.].   

We note that one or more of the authors are employed by a commercial company: Xegen, Gemenos and Praxis Genomics LLC

Reviewers' comments:

Reviewer's Responses to Questions

**Comments to the Author**

1. Is the manuscript technically sound, and do the data support the conclusions?

Reviewer #1: Partly

Reviewer #2: Partly

2. Has the statistical analysis been performed appropriately and rigorously? 

Reviewer #1: Yes

Reviewer #2: N/A

3. Have the authors made all data underlying the findings in their manuscript fully available?

Reviewer #1: Yes

Reviewer #2: Yes

4. Is the manuscript presented in an intelligible fashion and written in standard English?

Reviewer #1: Yes

Reviewer #2: Yes

5. Review Comments to the Author

Reviewer #1: The submitted manuscript, by Paganini et al, describes and validates a method to determine HLA-DRB1 and 9 blood group genotypes from whole genome Next Generation Sequencing (NGS) data. The described approach is technically sound, with appropriate description of the methods and of the important bioinformatic parameters in the results. Blood group genotyping is validated by comparing with previously-published SNaPshot data, and HLA-DRB1 typing is compared with amplicon-based monoallelic sequencing. Ethics approval is properly documented.

This research contribution demonstrates one of the advantages of employing NGS to predict red blood cell phenotypes: the capacity to also detect novel blood group alleles. The software employed includes physical phasing capabilities, allowing for unambiguous haplotype determination in many cases. The authors provide sufficient supplementary data for replication of this approach with other datasets, but the actual source NGS sequences do not appear to be publicly available.

Although the authors provide a thorough description of the bioinformatic pre-alignment process, there is no mention of the aligner used, the specific aligning parameters, and the human genome build employed in this study. Is this because the alignment is done by the PolyPheMe software as well?

The manuscript reports a potential novel weak FY allele, which on Table 1 is described as c.125A (p.Gly42Asp – the Fyb antigen) with c.298G>A (p.Ala100Thr) in cis. The presence of the Ala100Thr variant alone in a Fy(b) background was in fact reported previously in the literature (Olsson et al, BJH 1998, 103, 1184-1191) with no reported weakening of FY expression.

Four novel missense variants are reported by the study. For the Dombrock blood group the authors report p.Asp265Asn as a novel change; however, this actually corresponds to the known Do(a/b) antithetical antigens. This is however, one of the instances where the human genome build reference nucleotide does not match the ISBT reference table. In hg38, the reference nucleotide in chr12: 14840505 is a C, which leads to Asp265 (ART4 is coded in the minus strand). However, for immunohematologists and in the ISBT database, Asn265 is considered the reference.

The manuscript also lists p.Pro179Arg as a novel missense change in the JK blood group, however this variant is listed as part of JK*02N.13 in the most recent ISBT JK database (v6.0 01-MAR-2020 v2.0). Was this variant identified in cis with p.Asp280Asn (Jkb)? Was it identified without concomitant p.Met167Val, which is part of the definition of JK*02N.13?

The manuscript clearly reports the number of cases were heterozygosity was missed, and mentions that this was associated with low depth. It would be useful to provide the range of depth that was associated with this phenomenon, to aid in the determination of a minimum depth for this particular approach.

Minor revisions:

- Page 11, line, ‘targetingall’ is missing a space.

- Page 15, second paragraph, first line – the subject is singular (‘laboratory’) while the verb is plural (‘are’).

Reviewer #2: Manuscript number PONE-D-20-17434

The study aimed to validate red blood cell antigen typing from WGS data using a software validated for HLA typing.

Seventy – nine (79) samples, representing male Afghan volunteers, were used for the study. The samples had been tested by SNaPshot analysis. This SNaPshot analysis genotyped for SNVs associated with major blood group antigen polymorphisms for the nine blood groups analysed in this study.

The WGS data was compared to the previous published SNaPshot genotyping data.

WGS also revealed other variants including the presence of JK allelic variants associated with weak Jk antigen expression as well five SNVs in exonic regions for DO, IN and JK variants that have been described in Genebank but for which the blood group phenotype association is unknown. The limitation of the study is that no serology was performed and the authors note this in the discussion.

The study also revealed a number of intronic variants which could be informative in future population studies.

Major comments

1 The NGS data correlated with SNaPshot in defining alleles that were homozygous for the target SNV however there were five (102/107) incorrect typing results at the heterozygous positions. Of note, also a large number of positions (both homozygous and heterozygous) could not be called.

The authors should discuss whether the incorrect typing at heterozygous positions as well as this level of calls “not defined” is acceptable. The mean read depth of the genome was 11.8 which is not high and a reflection of the WGS approach used. Is this arguing for a need for a more targeted and efficient approach to obtain a higher read depth?

2 The approach of using the HLA software to interpret the blood group (Other than ABO) data appears novel although the study appears more a feasibility study than a validation study at this stage.

3 Please check the bi-allelic polymorphisms listed in the Material and Methods under Blood group genotyping e.g KEL, JK , etc. as a few are now not matching the most recently revised ISBT reference tables.

Other comments:

Results Page 11 Line 210 should the number be ‘148’ not ‘146’?

Please also check the ISBT terminology used e.g. use Jk (a+w ) phenotype

Discussion

First paragraph, last sentence: Please consider including an explanation for not including MNS because this was included in the study SNaPshot study.

The authors state in the discussion that the “results showed 99.5% of concordance for blood group polymorphisms” – please show how they arrived at this figure.

Editorial

Page 11 Line 189 ‘WGS-based typing targeting all…’ Space between targeting and all

Same paragraph – last sentence – do the authors mean ‘confirm or refute’…?

6. PLOS authors have the option to publish the peer review history of their article (what does this mean?). If published, this will include your full peer review and any attached files.

Reviewer #1: No

Reviewer #2: No

---

## [Author Response · Author response to Decision Letter 0]

21 Sep 2020

Dr. Di Cristofaro

UMR7268

Faculté de Médecine Timone

Marseille France

Santosh K. Patnaik, MD, PhD

Academic Editor

PLOS ONE

PONE-D-20-17434

BLOOD GROUP TYPING FROM WHOLE-GENOME SEQUENCING DATA

PLOS ONE

To the Editor and the Reviewers,

Thank you for having considered our manuscript for publication to PLOS ONE. We are grateful to both reviewers for their careful review. Accordingly, we are submitting a revised version of our manuscript that addresses the points raised during the review process. 

Please, find in this letter responses to each point raised by the editor and the reviewers. 

Editor comments:

> File naming and authors’ affiliations have been modified according to PLOS ONE style requirements.

2. Please provide additional details regarding participant consent. In the ethics statement in the Methods and online submission information, please clarify what type of consent you obtained (for instance, written or verbal, and if verbal, how it was documented and witnessed).

> Type of consent obtained from participants was clarified in the ethics statement in the Methods and online submission information as follows: « All samples were obtained from unrelated male Afghan volunteers after obtaining written informed consent. The study protocol was registered by the Ministere de l’Enseignement Superieur et de la Recherche in France (committee 208C06, decision AC-2008-232). Institutional review board Ministere de l’Enseignement Superieur et de la Recherche in France committee 208C06, (decision AC-2008-232) specifically approved this study. »

3. To comply with PLOS ONE submission guidelines, in your Methods section, please provide additional information regarding your statistical analyses. 

> Additional information regarding statistical analyses was added in the Methods section as follows : « Statistical analyses were performed with GRAPH PAD Prism 5 software (California USA, www.graphpad.com). Number of reads are presented as mean and range [min, max]. Differences among number of reads according to typing gene status were tested using Kruskal-Wallis one-way ANOVA for three values and Mann Whitney test for two values. Threshold for significance (alpha) was set at 0.05. »

4. Financial Disclosure section, Funding Statement and Competing Interests Statement modifications

>Funding, Financial disclosure, disclosure statement and authors’ contributions sections have been added to the manuscript as follows:

FUNDING 

No funding was received for this research. The funder provided support in the form of salaries for authors JP, PG and PN, but did not have any additional role in the study design, data collection and analysis, decision to publish, or preparation of the manuscript. The specific roles of these authors are detailed in the ‘author contributions’ section.

Financial Disclosure

The authors received no specific funding for this work. Authors Julien Paganini and Philippe Gouret are employed by a commercial company: Xegen, Gemenos, France. Author Peter L. Nagy is employed by a commercial company: Praxis Genomics LLC, Atlanta, Georgia, USA.

Disclosure Statement

The authors have no conflicts of interest to declare. Commercial affiliation of JP, PG and PN does not alter our adherence to all PLOS ONE policies on sharing data and materials.

Author Contributions

Conceptualization, Methodology, Visualization and Writing – Original Draft Preparation : JP and JDC; Data Curation and Formal Analysis : JP, PN, NR, PG and JDC ; Investigation Resources : CP, JC and JDC ; Project Administration : CP and JC ; Funding Acquisition : PN and JC ; Software : JP and PG ; Supervision and Validation: JC and CP ; Writing – Review & Editing : all authors

 

Reviewers’ comments:

The authors would like to thank both reviewers for their accurate review and kindness, their knowledge of blood groups have improved our manuscript. 

Reviewer #1 

Actual source NGS sequences do not appear to be publicly available.

> Authors agree with the reviewer. The reference of NGS has been added in the introduction and material section (Determination of the phylogenetic origins of the Árpád Dynasty based on Y chromosome sequencing of Béla the Third. Nagy PL, et al. Eur J Hum Genet. 2020 Jul 7. PMID: 32636469). The link for uploaded data has been added in the results section: “Sequencing data are available at http://www.ncbi.nlm.nih.gov/bioproject/662371”. This link will become public once the paper is published. A reviewer link is also normally created once the data set is fully processed as well.

Although the authors provide a thorough description of the bioinformatic pre-alignment process, there is no mention of the aligner used, the specific aligning parameters, and the human genome build employed in this study. Is this because the alignment is done by the PolyPheMe software as well?

> Authors agree with the reviewer. Additional information regarding alignment process was added in the Material and Methods section as follows: “WGS data were directly aligned to each gene as reference, no human genome was used for read mapping. Alignments were generated in PolyPheMe software with a Bowtie tool (Langmead et al. 2009; Langmead 2010).”

The manuscript reports a potential novel weak FY allele, which on Table S1 is described as c.125A (p.Gly42Asp – the Fyb antigen) with c.298G>A (p.Ala100Thr) in cis. The presence of the Ala100Thr variant alone in a Fy(b) background was in fact reported previously in the literature (Olsson et al, BJH 1998, 103, 1184-1191 9886340) with no reported weakening of FY expression.

> Authors thank the reviewer. Mention of weak FY phenotype was deleted concerning this FY allele with c.125A and c.298G>A in cis. Olsson et al. has been added as reference. 

Results and Tables were modified accordingly: “FY*02 allele associated with c.298G>A (p.Ala100Thr) was found in 18 samples (Olsson et al.; 1998).”

Discussion was modified as follows: “ We were able to type the JK*01W.01 allele (Jk (a+w)) (19) and the FY*02 allele associated with c.298G>A (p.Ala100Thr) (Olsson et al.; 1998).”

Four novel missense variants are reported by the study. For the Dombrock blood group the authors report p.Asp265Asn as a novel change; however, this actually corresponds to the known Do(a/b) antithetical antigens. This is however, one of the instances where the human genome build reference nucleotide does not match the ISBT reference table. In hg38, the reference nucleotide in chr12: 14840505 is a C, which leads to Asp265 (ART4 is coded in the minus strand). However, for immunohematologists and in the ISBT database, Asn265 is considered the reference.

The manuscript also lists p.Pro179Arg as a novel missense change in the JK blood group, however this variant is listed as part of JK*02N.13 in the most recent ISBT JK database (v6.0 01-MAR-2020 v2.0). Was this variant identified in cis with p.Asp280Asn (Jkb)? Was it identified without concomitant p.Met167Val, which is part of the definition of JK*02N.13?

> Authors thank the reviewer. We made a mistake when we used genebank sequences for translation. The coordinate was checked and corrected. DO and JK new polymorphisms induce no amino-acid change. Authors apologize for these mistakes. Mutations in IN gene were double checked. They are located after the codon stop (exon 9) in IN isoform 4 described in ISTB. This information has been added to the Supporting Table S6 legend.

Text and tables have been modified accordingly throughout the manuscript.

The manuscript clearly reports the number of cases were heterozygosity was missed, and mentions that this was associated with low depth. It would be useful to provide the range of depth that was associated with this phenomenon, to aid in the determination of a minimum depth for this particular approach.

> Authors thank the reviewer. Range of depth associated with incorrectly typed heterozygous samples was added in Supporting Table S3. Discussion has been modified accordingly: ”WGS data quality is assessed by the estimation of read depth. A former study conducted on WGS data established a minimum of 15x for RBC antigen typing in the clinical field (6, 14). Here, mean read depth of the genome was estimated at 11.8x [5.5x-18.4x] and read depth for each gene reached higher values. For each gene, typing resolution was significantly associated with the number of reads mapped on its sequence and ambiguous and incorrectly typing showed low numbers of reads corresponding to the missing allele and read depth equal to or below 15x. Our study thus confirms that RBC typing from WGS should be considered reliable with read depths strictly above 15x. To reach this goal, genome sequencing of one human (3Gb) should be analyzed with at least 45 Gb of data, here mean data was 34 Gb [16-53]”. Also, this sentence has been added in the abstract: “Our study confirms that RBC typing from WGS should be considered reliable with read depths strictly above 15x.”

Minor revisions:

- Page 11, line, ‘targetingall’ is missing a space.

- Page 15, second paragraph, first line – the subject is singular (‘laboratory’) while the verb is plural (‘are’).

> Authors thank the reviewer, typos have been corrected.

 

Reviewer #2

Major comments

1 The authors should discuss whether the incorrect typing at heterozygous positions as well as this level of calls “not defined” is acceptable. The mean read depth of the genome was 11.8 which is not high and a reflection of the WGS approach used. Is this arguing for a need for a more targeted and efficient approach to obtain a higher read depth?

> Authors agree with the reviewer, the corresponding paragraph has been modified as follows: ” WGS data quality is assessed by the estimation of read depth. A former study conducted on WGS data established a minimum of 15x for RBC antigen typing in the clinical field (6, 14). Here, mean read depth of the genome was estimated at 11.8x [5.5x-18.4x] and read depth for each gene reached higher values. For each gene, typing resolution was significantly associated with the number of reads mapped on its sequence and ambiguous and incorrectly typing showed low numbers of reads corresponding to the missing allele and read depth equal to or below 15x. Our study thus confirms that RBC typing from WGS should be considered reliable with read depths strictly above 15x. To reach this goal, genome sequencing of one human (3Gb) should be analyzed with at least 45 Gb of data, here mean data was 34 Gb [16-53].”

This sentence has been added in the abstract: “Our study confirms that RBC typing from WGS should be considered reliable with read depths strictly above 15x.”

2 The approach of using the HLA software to interpret the blood group (Other than ABO) data appears novel although the study appears more a feasibility study than a validation study at this stage.

> Authors agree with the reviewer, abstract and introduction have been modified accordingly.

3 Please check the bi-allelic polymorphisms listed in the Material and Methods under Blood group genotyping e.g KEL, JK , etc. as a few are now not matching the most recently revised ISBT reference tables.

> Authors agree with the reviewer, ISBT Terminology has been checked.

Other comments:

Results Page 11 Line 210 should the number be ‘148’ not ‘146’?

Please also check the ISBT terminology used e.g. use Jk (a+w ) phenotype. 

> Authors agree with the reviewer, the number has been corrected, also abstract, results and discussion sections have been modified accordingly: “Our results showed 93% of concordance for blood group polymorphisms and 91% for HLA-DRB1.”

ISBT Terminology has been checked.

Discussion

First paragraph, last sentence: Please consider including an explanation for not including MNS because this was included in the study SNaPshot study.

> Authors agree with the reviewer. As for RHD/RHCE system, MNS could not be included because of the existence of hybrids. The sentence has been modified accordingly: “Whereas targeted strategies, such as PCR followed by sequencing or SnaPshot, circumvent specificity issues of genes with structural changes and hybrids such as RHCE/RHD and GPA/GPB ; their analysis from WGS data requires specific bioinformatic approaches including CNV (copy number variation) analysis. Therefore, such systems were not included in this study.”

The authors state in the discussion that the “results showed 99.5% of concordance for blood group polymorphisms” – please show how they arrived at this figure.

> WGS analyses allow identification of 972 SNPs; 967 SNP were correctly identified (i.e. 99.5%). 93.4% of concordance was obtained when taking into account ambiguous typing. The sentence in the discussion was modified as follows: “Our results showed that blood group typing deduced from WGS were correct at 99.5% compared to SNaPshot analysis (967 SNP correctly identified out of 972 typed); 93% of concordance was obtained when taking into account ambiguous typing. In a clinical or research context however, ambiguous RBC results need to be reanalyzed. HLA-DRB1 typing from WGS showed 91% of concordance with those obtained by amplicon-based monoallelic sequencing.”

Editorial

Page 11 Line 189 ‘WGS-based typing targeting all…’ Space between targeting and all

Same paragraph – last sentence – do the authors mean ‘confirm or refute’…?

> Authors thank the reviewer, modifications have been made.

The authors would like to thank reviewers for their critical review, we feel that our manuscript has been improved. We look forward to the editor and reviewers’ comments and the editorial board’s decision. 

Please do not hesitate to contact us if you require any further information.

Yours sincerely,

---

## [Decision Letter · Decision Letter 1]

21 Oct 2020

PONE-D-20-17434R1

BLOOD GROUP TYPING FROM WHOLE-GENOME SEQUENCING DATA

PLOS ONE

Dear Dr. Di Cristofaro,

Thank you for submitting your revised manuscript to PLOS ONE. It has now been examined by one of the two referees who had reviewed the original submission.

Based on the new review, I am making a decision for 'major revision' and requesting you to submit a revised version of the manuscript that addresses the points raised by the reviewer.

We look forward to receiving your revised manuscript.

Kind regards,

Santosh K. Patnaik, MD, PhD

Academic Editor

PLOS ONE

Reviewers' comments:

Reviewer's Responses to Questions

**Comments to the Author**

1. If the authors have adequately addressed your comments raised in a previous round of review and you feel that this manuscript is now acceptable for publication, you may indicate that here to bypass the “Comments to the Author” section, enter your conflict of interest statement in the “Confidential to Editor” section, and submit your "Accept" recommendation.

Reviewer #2: (No Response)

2. Is the manuscript technically sound, and do the data support the conclusions?

Reviewer #2: Partly

3. Has the statistical analysis been performed appropriately and rigorously? 

Reviewer #2: N/A

4. Have the authors made all data underlying the findings in their manuscript fully available?

Reviewer #2: Yes

5. Is the manuscript presented in an intelligible fashion and written in standard English?

Reviewer #2: Yes

6. Review Comments to the Author

Reviewer #2: PLOS ONE-D-20-1743R1

The authors have clarified some of the queries with regard to Incorrect Typing and with regard to Blood Group Nomenclature however anomalies still remain in both regards.

Abstract the Conclusion:

Page 2 Line 31 – 32,

The authors should reconsider the conclusion which is too general for the results presented in this study.

The stated aim includes estimating the feasibility of RBC antigen typing from WGS data.

Table 2 shows the authors find incorrect typing results for 5 samples and correct results for 102 samples with heterozygous calls. They also find a large number of not determined (unresolved) results with 53 homozygous samples and 10 heterozygous not resolved.

The authors elsewhere show that the read depth is insufficient at 11.8 which supports previous findings that read depth needs to be above 15.

The conclusion is that RBC antigen typing is feasible however improvements in read depth are needed to improve the accuracy in typing for SNV polymorphisms.

The paper does show the potential for WGS in detecting other alleles, such as the weak JK alleles, with potential to be used to screen for rare donors.

Nomenclature:

Materials and Methods:

Page 5 line 84 & 85 the bracketed list gives the internationally HUGO blood group gene names with the Blood Group System symbols in brackets. However the gene names given for FY, IN, CO DI and LW are incorrect. The gene names for these Blood Group Systems are ACKR1 (FY), CD44 (IN), AQP1 (CO), SLC4A1 (DI) and ICAM4 (LW).

Reference http://www.isbtweb.org/fileadmin/user_upload/Table_of_blood_group_systems_v6.0_6th_August_2019.pdf

Page 5 Line 87 to 89: After “Fourteen SNPs were analysed corresponding to bi-allelic polymorphisms” the authors show the following amino acid changes (phenotypes). The Duffy needs editing to match ISBT Tables as follows:

Page 5 Line 89: The phenotype for (FY p.Arg89Cys) is Fya+w, and the phenotype for the FY promoter change (p.0) with the FYc-67T>C change is (Fy(a-b-) erythroid cells only)

Presentation of data in Table 2:

There are ambiguities in the presentation of data in Table 2.

For example for FY the third column and fourth column indicate that the FY*01N.01 allele showed Homozygous Correct Typing for 64 cases with 8 not defined. This is generally a rare allele.

The next line shows for the FY*02 allele (the alternate for the FY*01) that 32 were Homozygous. Yet there are only 79 individuals in the study and these numbers as written exceed the number in the study (64+ 32). This is not counting the 26 Heterozygous cases and 7 incorrect or undefined.

The next line indicates for FY*0W.01 there are a further 61 cases homozygous (now there are 64 + 32 + 61).

Again in Table 2

The authors also report 70 Homozygous for IN*01 which is indeed a very rare allele and should, if confirmed, be commented on.

References

Introduction Line 46: Suggest include the papers by Lane et al with (1,2) as well as in the next paragraph.

Minor Edits:

Introduction: Line 3: Research and CE-environment labs – suggest Research and Clinical laboratories working within regulatory approved frameworks e.g. Council of Europe (CE) .

7. PLOS authors have the option to publish the peer review history of their article (what does this mean?). If published, this will include your full peer review and any attached files.

Reviewer #2: No

---

## [Author Response · Author response to Decision Letter 1]

26 Oct 2020

To the Editor and the Reviewer,

Thank you for having considered our revised manuscript for publication to PLOS ONE. We are grateful to reviewer for her/his second review. Accordingly, we are submitting a revised version of our manuscript. 

Please, find in this letter responses to each point raised by the reviewer. 

*******

Reviewer #2

The authors have clarified some of the queries with regard to Incorrect Typing and with regard to Blood Group Nomenclature however anomalies still remain in both regards.

Abstract the Conclusion:Page 2 Line 31 – 32, The authors should reconsider the conclusion which is too general for the results presented in this study. The stated aim includes estimating the feasibility of RBC antigen typing from WGS data.

Table 2 shows the authors find incorrect typing results for 5 samples and correct results for 102 samples with heterozygous calls. They also find a large number of not determined (unresolved) results with 53 homozygous samples and 10 heterozygous not resolved.

The authors elsewhere show that the read depth is insufficient at 11.8 which supports previous findings that read depth needs to be above 15.

The conclusion is that RBC antigen typing is feasible however improvements in read depth are needed to improve the accuracy in typing for SNV polymorphisms.

The paper does show the potential for WGS in detecting other alleles, such as the weak JK alleles, with potential to be used to screen for rare donors.

> Authors agree with the reviewer, the abstract has been modified: “Incorrect typing and unresolved results confirm that WGS should be considered reliable with read depths strictly above 15x. Our results supported that RBC antigen typing from WGS is feasible but requires improvements in read depth for SNV polymorphisms typing accuracy. We also showed the potential for WGS in screening donors with rare blood antigens, such as weak JK alleles. The development of WGS analysis in immunogenetics laboratories would offer personalized care in the management of RBC disorders.

Nomenclature:

Materials and Methods:

Page 5 line 84 & 85 the bracketed list gives the internationally HUGO blood group gene names with the Blood Group System symbols in brackets. However the gene names given for FY, IN, CO DI and LW are incorrect. The gene names for these Blood Group Systems are ACKR1 (FY), CD44 (IN), AQP1 (CO), SLC4A1 (DI) and ICAM4 (LW).

Reference http://www.isbtweb.org/fileadmin/user_upload/Table_of_blood_group_systems_v6.0_6th_August_2019.pdf

> Authors thank the reviewer, gene names have been corrected.

Page 5 Line 87 to 89: After “Fourteen SNPs were analysed corresponding to bi-allelic polymorphisms” the authors show the following amino acid changes (phenotypes). The Duffy needs editing to match ISBT Tables as follows: Page 5 Line 89: The phenotype for (FY p.Arg89Cys) is Fya+w, and the phenotype for the FY promoter change (p.0) with the FYc-67T>C change is (Fy(a-b-) erythroid cells only)

> Authors thank the reviewer, phenotypes have been corrected.

Presentation of data in Table 2:

There are ambiguities in the presentation of data in Table 2. For example for FY the third column and fourth column indicate that the FY*01N.01 allele showed Homozygous Correct Typing for 64 cases with 8 not defined. This is generally a rare allele. The next line shows for the FY*02 allele (the alternate for the FY*01) that 32 were Homozygous. Yet there are only 79 individuals in the study and these numbers as written exceed the number in the study (64+ 32). This is not counting the 26 Heterozygous cases and 7 incorrect or undefined. The next line indicates for FY*0W.01 there are a further 61 cases homozygous (now there are 64 + 32 + 61). Again in Table 2, The authors also report 70 Homozygous for IN*01 which is indeed a very rare allele and should, if confirmed, be commented on.

> Authors thank the reviewer, ambiguities in table 2 was due to allele names instead of polymorphisms. No sample was typed FY*01N.01; among the 72 homozygous samples (wild type as shown in Table 1) at position ACKR1 (-67T>C), 64 samples were correctly defined and 8 samples were not defined. Among the 35 samples homozygous at position ACKR1 (125G>A), whether wild type or mutated, 32 were correctly defined and 3 were not defined. Among the 30 samples heterozygous at the same position, 26 samples were correctly typed, 1 sample was not correctly typed and 3 samples were not defined. No sample was typed IN*01. Among the 72 homozygous samples (wild type as shown in Table 1) at position CD44 (137G>C), 70 samples were correctly defined and 2 were not defined.

Allele names have been replaced with polymorphisms in Table 2.

References

Introduction Line 46: Suggest include the papers by Lane et al with (1,2) as well as in the next paragraph.

> Authors agree with the reviewer, reference has been added

Minor Edits:

Introduction: Line 3: Research and CE-environment labs – suggest Research and Clinical laboratories working within regulatory approved frameworks e.g. Council of Europe (CE).

> Authors agree with the reviewer, text has been modified accordingly.

The authors would like to thank the reviewer. We look forward to the editor and reviewer’s comments and the editorial board’s decision. 

Please do not hesitate to contact us if you require any further information.

Yours sincerely,

---

## [Editor Report · Decision Letter 2]

28 Oct 2020

BLOOD GROUP TYPING FROM WHOLE-GENOME SEQUENCING DATA

PONE-D-20-17434R2

Dear Dr. Di Cristofaro,

Thank you for submitting the second revised version of your manuscript for our appraisal. The concerns raised by Referee #2 in the last review have been appropriately addressed with the revision. The manuscript is therefore scientifically suitable for publication and it will be formally accepted for publication once it meets all outstanding technical requirements.

Kind regards,

Santosh K. Patnaik, MD, PhD

Academic Editor

PLOS ONE
---

## [Editor Report · Acceptance letter]

4 Nov 2020

PONE-D-20-17434R2 

BLOOD GROUP TYPING FROM WHOLE-GENOME SEQUENCING DATA 

Dear Dr. Di Cristofaro:

I'm pleased to inform you that your manuscript has been deemed suitable for publication in PLOS ONE. Congratulations! Your manuscript is now with our production department. 

Kind regards, 

on behalf of

Dr. Santosh K. Patnaik 

Academic Editor

PLOS ONE